# Diet Preference, Feed Efficiency and Expression of the Sodium-Dependent Glucose Transporter Isoform 1 and Sweet Taste Receptors in the Jejunum of Lambs Supplemented with Different Flavours

**DOI:** 10.3390/ani13081417

**Published:** 2023-04-20

**Authors:** Felista Mwangi, Areen Dallasheh, Mugagga Kalyesubula, Naama Reicher, Chris Sabastian, Sameer J. Mabjeesh

**Affiliations:** 1Department of Animal Science, The Robert H. Smith Faculty of Agriculture, Food and Environment, The Hebrew University of Jerusalem, Rehovot 760001, Israel; felista.mwangi@mail.huji.ac.il (F.M.); areen.dallasheh@mail.huji.ac.il (A.D.); mugagga.kalyesubula@mail.huji.ac.il (M.K.); naama.reicher@mail.huji.ac.il (N.R.); sabastian.c@gmail.com (C.S.); 2Animal Genetics and Nutrition, Veterinary Sciences Discipline, College of Public Health, Medical and Veterinary Sciences, Division of Tropical Health and Medicine, James Cook University, Townsville, QLD 4811, Australia

**Keywords:** taste receptor family 1 member 2, taste receptor family 1 member 3, sucram, capsicum, feed intake, sheep

## Abstract

**Simple Summary:**

Flavours improve the sensory characteristics of the feed to increase voluntary intake of novel by-products and monotonous feeds, and improve feed efficiency, resulting in improved body weight gain. However, there is a scarcity of studies on the effect of flavour on the preference and feed efficiency in sheep raised in intensive farming systems post-weaning. Therefore, this study investigated the flavour preference in lambs and the effect of flavours on feed efficiency, sweet taste receptors and sodium-glucose linked transporter 1 gene expression in the small intestines. Eight lambs were offered 16 different commercial flavours in a grain mixture of rolled barley and ground corn to examine the preference. In the metabolic study, the lambs were randomly assigned to four treatments (sucram, capsicum, a mix containing sucram and capsicum at 1:1 ratio and no flavour for control) in a 4 × 2 cross-over design. The lambs showed a preference for capsicum, sucram and milky flavours, and disliked the liquid orange flavour. Flavours did not improve feed intake in the metabolic study, but capsicum increased the average daily weight gain per metabolic body weight. These findings indicate that flavours can be used to motivate feed acceptance and improve the weight gain in lambs.

**Abstract:**

This study investigated the effect of dietary flavour supplements on the preference, feed efficiency and expression of the sweet taste receptor family 1 members 2 and 3 (*T1R2* + *T1R3*), and sodium-glucose linked transporter 1 (*SGLT1*) genes in the lambs’ small intestines. Eight, five-month-old, Israeli crossbred Assaf lambs were offered 16 different non-nutritive commercial flavours in rolled barley and ground corn. Capsicum and sucram were the most preferred non-aroma flavours (*p* = 0.020), while milky (*p* < 0.001) was the most preferred powder-aroma flavour. For the metabolic and relative gene expression study, eight lambs were randomly assigned to either sucram, capsicum, a mix containing sucram and capsicum at 1:1 ratio or no flavour for control in a 4 × 2 cross-over design. The total collection of urine (females only), faeces and refusals was carried out, and *T1R2*, *T1R3* and *SGLT1* relative gene expression evaluated from the proximal jejunum biopsies. Flavour had no significant effect on the feed intake (*p* = 0.934), but capsicum increased the average daily weight gain per metabolic body weight (*p* = 0.049). The *T1R3* gene was expressed highest in the mix treatment (1.7; *p* = 0.005). Collectively, our findings indicate that flavours can be used to motivate feed acceptance and improve the weight gain in lambs.

## 1. Introduction

The sensory component of feed plays a major role in the palatability complex to control ingestive behaviour, reinforcement in learning situations and the onset of specific appetites [1]. In the current intensive feeding systems, animals consume monotonous diets. Studies from rats and humans have shown that eating the same feed may lead to a reduced feed intake and slow growth rate [2]. Moreover, animals suffer low voluntary feed intake, corresponding with poor growth after weaning, which poses major limitations to the enhanced efficiency of production [3,4,5].

Flavours can improve the sensory characteristics of the feed, increase voluntary intake of by-product feeds [1,6], restore motivation to eat by enhancing feed acceptability and permit feed formulation changes without affecting feed intake patterns [7,8]. Ruminants are reported to eat unfamiliar feed more willingly when they are associated with a familiar flavour or odour, and avoid the feed when presented with a novel odour. Preference of some flavours to others is also reported [9]. However, no consensus has been reached on the effect of flavouring ruminant feed on preference, feed intake or feed efficiency. For instance, Montoro and colleagues [10] reported that flavouring calves’ starter had no significant effect on feed intake for calves with average or above-average appetite, but it stimulated intake for calves with low appetite (less than 600 g/day dry matter intake (DMI)). In dairy cows fed seven different commercial flavours, they were reported to prefer fenugreek and vanilla in a cafeteria experiment [11]. In another study, young goats exposed to five flavours preferred orange and lemon, increasing the DMI by 35% compared to the control, while grapefruit flavour reduced the DMI by 14% (Mabjeesh unpublished data). Furthermore, feeding differently flavoured feed to post-pubertal heifers was reported to alter short-term preferences [12]. However, no studies have evaluated the preference of commercially available liquid and powdered flavours for lambs.

Furthermore, the inclusion of sweeteners or capsaicin-based flavours in diets have been reported to influence feed efficiency and nutrient metabolism [13,14,15,16]. Dietary sweeteners bind to the sweet taste receptors and induce an intracellular transduction cascade that activates the sodium-glucose linked transporter 1 (*SGLT1*), and subsequently, the glucose uptake [17,18]. Moreover, the capsaicin receptor (vanilloid receptor subtype 1) was reported to be co-localized with the sweet taste receptors in rat circumvallate papillae [19], suggesting that capsaicin may influence the sweet taste transduction pathways. Gu et al. [20] reported that oral exposure to capsaicin increased consumption of sucrose and saccharin-sweetened solutions in rats. Furthermore, capsaicin has been reported to increase the apparent total-tract feed digestibility in dairy cows [16], but the underlying mechanisms are not well understood. Moreover, there is a paucity of studies evaluating the effect of dietary sweeteners and capsaicin on feed digestibility in sheep. Since gene expression may indicate protein abundance [21], there is need for studies evaluating the expression of the sweet taste receptors and glucose transporters to unravel the underlying mechanism of sweet flavours and capsaicin on feed efficiency.

Therefore, this study aimed to characterize the effect of flavour compounds in the diet of lambs on preference and feed efficiency in intensive farming, and to use the basic knowledge at the cellular level to examine the relationship between flavours and the absorption of energetic compounds in the small intestines. We hypothesized that lambs prefer some flavours over others, and supplementing feed with their preferred flavours after weaning increase feed acceptance, intestinal expression of sweet taste receptors and monosaccharide transporters that lead to increased feed efficiency.

## 2. Materials and Methods

The Hebrew University (Rehovot, Israel) Institutional Animal Care and Use Committee reviewed and approved all procedures in this study (protocol code AG-15086).

### 2.1. Preference Study

To evaluate flavour preference in sheep, eight five-month-old Israeli crossbred Assaf lambs (four females and four males) weighing 41.0 ± 4.8 kg were purchased from a commercial farm. The lambs were fed concentrated pellets and a total mixed ration at 60% and 40% (on DM basis; Table 1), respectively, to meet their metabolic needs [22]. The lambs were housed in metabolic cages (105 cm by 60 cm) in climate-controlled rooms with 14 h of light and 10 h of dark cycles, 22 ± 2 °C temperature and 55–60% humidity. Lights were turned on at 0600 and off at 2000 h. Water was accessible ad libitum while the diet was divided into 12 equal meal portions dispensed by automatic feeders every two hours (ANKOM Technology, Macedon, NY, USA). Feed quantities were adjusted every day to ensure at least 5% refusals. Lambs were allowed a seven day adaptation period to the basal diet and experimental facility. The preference test was then conducted according to the modified protocol of Harper et al. [11] using 16 different non-nutritive commercial flavours: three liquid flavours from Frutarom^®^ (Haifa, Israel) and 13 powdered flavours from Pancosma^®^ (Geneva, Switzerland). The liquid flavours were concentrated vanilla (2233874610), concentrated orange (2222351010) and pineapple (2222307810). The powder flavours were classified into two categories: aromas and non-aromas. Aroma flavours included vanilla shake (Pan Vanilla Shake A60-6416; flavour commercial name and flavour code, respectively), milky (Pan-TEK Milky A60-3510), vanifen (Pan VaniFen C60-3124), red summer fruits (Pan Red Summer Fruits A60-3132), molasses (Pan TEK Molasses Extra A60-3423), anis (Pan Anis A61-3041), citrus (Covotek Citrus 6573), orange (Pan Juicy Orange A60-3134), honey vanilla Jasmin (6659) and fenugreek (6571). Non-aroma flavours were capsicum (Xtract CAPS XL X60-7035), nexulin (Nexulin FM N60-3302) and sucram (Sucram C-150 6834). Flavour-concentrate mixes were prepared at the beginning of the experiment and stored at 4 °C.

One hour after the morning meal, each lamb was offered two different flavours mixed with 100 g of grains (rolled barley and ground corn at 1:1 ratio as fed) in separate buckets for 5 min. The flavour inclusion rate was as follows: 150 g/ton for non-aromas (capsicum, nexulin and sucram), 300 g/ton for powder aromas and 0.5% on DM basis for the liquid aromas, according to manufacturer’s recommendations. An unflavoured grain mixture was used as the control and offered at a random time similar to the flavoured treatments. The treatments were replicated twice, once in two days, to prevent carry-over effects, and all the lambs had access to all of the flavours. Each time, the relative location of the buckets in the feeders were changed when the same flavours were tested. Lambs were not adapted to any of the flavours before the experiment started. Preference was expressed as g consumed within the 5 min test.

### 2.2. Feed Intake, Performance, Nutrient Apparent Digestibility and Biopsy Sampling of Intestinal Tissue

To examine the effect of the flavours on feed intake, weight gain, apparent feed digestibility, energy balance and relative gene expression in the small intestines, a metabolic experiment was conducted with eight lambs (4 males and 4 females) using the sucram and capsicum flavours. Sucram and capsicum were selected because their preference was similar to the control during the preference test, and they are reported to influence feed efficiency in mammals [22,23]. The lambs (one male and one female) were assigned to either sucram, capsicum, a mix (sucram and capsicum at 1:1 ratio) or no flavour control supplemented in the feed. The feed consisted of a total mixed ration, whole corn grain and pellets in a 50:25:25 ratio (weight as is), respectively, formulated to ensure 15.7% CP and 2.68 Mcal/kg metabolizable energy (ME) on DM basis (Table 1). Every morning, feed and flavours were weighed and mixed thoroughly by hand for two minutes, partitioned into 12 equal meal portions and loaded into the automatic feeders. Feed quantities were adjusted every day to ensure at least 5% refusals.

Following 14 days of exposure to the flavours, the total collection of urine, faeces and refusals was carried out for five days. Urine was collected using catheters (in females only; Foley catheter 16 Fr.; Degania Silicone Ltd., Hatzor HaGlili, Israel) into 1500 mL urine bags with 10 mL of 20% sulphuric acid as a preservative while faeces were collected on a net fitted under the metabolic cages. Total collected urine and faeces were weighed and stored at −20 °C until chemical analysis. The treatments were switched at the end of the first period and the experiment repeated for the second period. Sheep were weighed at the beginning and at the end of each period of the experiment to determine weight gain (Scales Galore, Brooklyn, NY, USA), and daily weight gain was expressed in grams per metabolic body weight (g/Kg BW^0.75^) to control for baseline weight variability between periods.

At the end of each total collection period, the feed was withdrawn from the lambs for 24 h in preparation for the biopsy procedure. The procedure was performed under full anaesthesia, as follows. Lambs were sedated with Xylazine (0.05 mg/kg body weight (BW); Sigma-Aldrich, Rehovot, Israel). Anaesthesia was then initiated by intravenous injection of ketamine (2.2 mg/kg BW; Chanelle, Berkshire, UK) and Assival to affect (TEVA Pharmaceutical Industries Ltd., Petach Tikva, Israel). Anaesthesia was maintained with isoflurane gas (Sigma-Aldrich, Rehovot, Israel) utilizing an anaesthetic machine (Vetland Medical Sales and Services LLc, Louisville, KY, USA) at 2%. For venous catheterisation, the right arm was shaved, scrubbed thoroughly with septal scrub and rinsed with 70% ethanol. A longitudinal incision was made on the right side of the abdominal flank after surgical scrub and preparation. Approximately 3 cm of intestinal tissue was taken from the proximal jejunum (10–15 cm post the duodenal loop) using the full-thickness biopsy technique (resection and anastomosis). Biopsied tissue samples were washed with isotonic saline, immediately snap-frozen in liquid nitrogen and stored at −80 °C awaiting gene expression analysis.

All of the lambs were healthy throughout the experiment, except one male lamb that died after the first period of the experiment. As a result, data for the first period of the experiment consisted of eight lambs, while the second period consisted of seven lambs.

### 2.3. Chemical Analysis

To examine the effect of flavours on feed component intake, digestibility and energy balance, feed, faeces and urine were analysed for chemical composition, as follows. Refusals and faeces samples were left at room temperature overnight to thaw. The five-day collections for each lamb were then composited and mixed manually to homogenize. Faeces were then homogenized further using a blender (Magimix SAS, Vincennes, France). Samples of feed, faeces and refusals were dried at 60 °C in a circulated forced oven for 48 h and ground to pass through a 2 mm screen using a Wiley knife mill (Thomas Scientific, Swedesboro, NJ, USA) before analysis.

The DM content of the oven-dried samples was determined by drying at 105 °C for 12 h [24]. Ash content was determined by combustion in a muffle furnace at 600 °C for 3 h, while organic matter (OM) was calculated as the difference between DM and ash contents. Neutral detergent fibre (NDF) and acid detergent fibre (ADF) contents were determined using Ankom fibre analyzer (Ankom Technology, Macedon, NY, USA) [25] with heat-stable alpha amylase (Sigma-Aldrich, Rehovot, Israel) used in the NDF procedure. Crude protein (CP = N × 6.25) was determined using the Kjeltec machine (Soltek Analytics Ltd., Modi’in, Israel) for N analysis of feedstuffs according to the Kjeldhal method [24]. Gross energy (GE) content was determined by calorimetry using a bomb calorimeter (Parr 6100 calorimeter, Parr Instrument Company, Moline, IL, USA), while digestible energy (DE), ME, combustible gas energy (GasE), heat increment energy (HiE), net energy (NE) and apparent nutrient digestibility were estimated, as described by the National Research Council [22,26].

### 2.4. Determination of mRNA Abundance by Real-Time Polymerase Chain Reaction

To examine the effect of supplementing diet with flavours on intestinal sweet taste receptors and SGLT1, total RNA was isolated from the intestinal samples using TRIzol Reagent (Life Technologies, Carlsbad, CA, USA) according to the manufacturer’s protocol. Total RNA concentration was determined using NanoDrop ND-1000 spectrophotometer (Thermo Fisher Scientific, Waltham, MA, USA) before dilution to 200 ng/µL. A 1.0 µg of the total RNA from each tissue was reverse transcribed to cDNA using qPCRBIO cDNA synthesis kit (PCR Biosystems Inc., Wayne, PA, USA) according to the manufacturer’s protocol in a T100™ Bio-Rad Instrument. The PCR products were authenticated by electrophoresis in 1.5% agarose gel.

To assess the relative mRNA abundance of taste receptor family 1 member 2 (*T1R2*), taste receptor family 1 member 3 (*T1R3*), sodium-dependent glucose transporter isoform *1* (*SGLT1*) and the reference genes glucose-6-phosphate dehydrogenase (*G6PDH*), beta-2 microglobulin (*B2M*) and glyceraldehyde 3-phosphate dehydrogenase (*GAPDH*), gene-specific primers were designed with the aid of Primer-BLAST [27] according to the published cDNA sequences (Table 2) and purchased from Sigma-Aldrich (Park Rabin, Rehovot, Israel). The qPCR was performed using a Roche Lightcycler 96 (Roche Molecular Systems, Inc., Pleasanton, CA, USA). The reaction mix contained 10 µL Platinum SYBR Green qPCR supermix-UDG (Thermo Fisher Scientific, Rhenium, Israel), 1 µL of (4 µM) forward and reverse primers each, 3 µL of template (cDNA) diluted at 1:10 and 5 µL ultra-pure water, to make a total reaction volume of 20 µL. All reactions were performed in triplicates in a 96-well PCR microplate (Axygen Scientific, Inc., Union City, CA, USA) covered with optically clear sealing film (Axygen Scientific, Inc., CA, USA). The reaction conditions were: (1) pre-incubation at 50 °C for 120 s and 95 °C for 120 s, (2) three step amplification characterised by 45 cycles of 95 °C for 10 s, 60 °C for 30 s and 72 °C for 10 s, and (3) melting at 95 °C for 60 s, 65 °C for 60 s and 97 °C for 1 s. A no-template control was included to control for false positives, while positive control was included to control for false negatives. Cycle threshold (Ct) values from Roche LightCycler 96 program were used to calculate the relative gene expression using the delta-delta Ct method [28].

### 2.5. Statistical Analysis

Data were analysed using the SAS in JMP Pro version 13 (SAS Institute Inc., Cary, NC, USA), and graphs were plotted using the GraphPad Prism software version 9.4.1 (GraphPad Software, San Diego, CA, USA). To assess the effect of flavour on preference in lambs, consumption data were analysed using a repeated measures analysis of variance procedure. The model included flavour, run (first or subsequent exposure to the flavour), sex, flavour-by-run interaction and flavour-by-sex interaction as the fixed effects, while the individual lambs’ nested in sex, lamb-by-run and lamb-by-flavour interactions were included as the random effects. The effects of supplementing feed with sucram and capsicum on feed intake, digestibility, body weight gain, energy balance and gene expression were analysed by a mixed-effects model fitted with the restricted maximum likelihood method. Dependent variables included the fixed effects of treatment (flavour), sex, period and period by treatment interaction. The individual lambs nested in sex were included as the random effect. The urine energy data were analysed without the effect of sex because urine samples were collected from females only. Differences were considered significant at *p* ≤ 0.05 and tendencies at *p* < 0.1. Where significant, Tukey’s honestly significant difference test was used to separate the means.

## 3. Results

### 3.1. Flavour Preference

The flavour preference data are represented in Figure 1 and Figure 2. The effect of run, sex and sex by flavour interactions on flavour preference were not significant (*p* > 0.05); hence they were not tabulated. Lambs showed a preference for vanilla, unflavoured (control) and pineapple than orange flavoured grain mixture when offered liquid aroma flavours (Figure 1A; *p* < 0.001). Capsicum and sucram were preferred over nexulin in non-aroma flavours (Figure 1B; *p* = 0.020), while milky flavour was preferred over the red summer fruits, citrus, honey vanilla jasmin, vanilla shake and fenugreek in powdered aromas (Figure 1C; *p* < 0.001). Consumption of flavoured grain mixture did not differ from the control except for the liquid orange flavour that reduced consumption by 47%. Flavour by run interaction was observed for the non-aroma flavours (Figure 2A; *p* = 0.023) and powder-aroma flavours (Figure 2B; *p* < 0.001), where consumption was equal to or higher than during the second exposure compared to the initial exposure, except for the sucram, milky, molasses and vanifen that were consumed less during the second exposure.

### 3.2. Feed Intake, Performance and Nutrient Apparent Digestibility

The feed intake, weight gain and apparent digestibility of feed components are presented in Table 3 and Table 4 and in Figure 3 and Figure 4. There was no flavour effect on DM, OM, CP, NDF, ADF, hemicellulose and EE intake (Table 3; *p* ≥ 0.893). However, the capsicum flavour increased the average daily weight gain by 3.1 and 5.9 g/Kg BW^0.75^ compared to the control and the mixed capsicum and sucram flavours, respectively (Figure 4A; *p* = 0.049). The digestibility of DM, OM, CP, NDF, ADF and hemicellulose did not differ between treatments (*p* ≥ 0.265), but the EE digestibility tended to be higher in capsicum and lower in the sucram fed lambs (Table 4; *p* = 0.083). The males were observed to have a higher feed intake (Figure 3; *p* ≤ 0.003) and gained 3.8 g more weight per Kg BW^0.75^ (Figure 4B; *p* = 0.024) compared to the females, but diet digestibility did not differ between male and female lambs (*p* > 0.05).

### 3.3. Energy Balance

Feeding lambs with unflavoured diet or flavoured with sucram, capsicum or both did not have a significant effect on the GEI, DE, FE, UE, GasE, ME, HiE and NE (Table 5; *p* ≥ 0.723). However, the daily GEI, DE, FE, GasE and HiE were higher in males compared to the females (Figure 5; *p* ≤ 0.040). The males consumed 77.9 Kcal/Kg BW^0.75^ more (*p* = 0.004) and lost 20.7 Kcal/Kg BW^0.75^ higher than the females as faecal energy (*p* = 0.040).

### 3.4. T1R2, T1R3 and SGLT1 Gene Expression

Lambs expressed all the measured genes (*T1R2*, *T1R3* and *SGLT1*) in the proximal jejunum (Figure 6). A mix of sucram and capsicum flavours increased the T1R3 relative gene expression compared to each flavour fed alone or the unflavoured (*p* = 0.005). However, the T1R2 (*p* = 0.193) and SGLT1 (*p* = 0.378) gene expression did not differ between treatments.

## 4. Discussion

### 4.1. Flavour Preference

Lambs in the present study preferred vanilla over orange and pineapple flavours. In agreement with our findings, vanilla was the most preferred flavour when cows were offered anise, orange, vanilla, honey, thyme and molasses in a cafeteria experiment reported by Harper et al. [11]. In another study, ewes offered a eucalyptus, mint, orange and oregano-flavoured commercial diet preferred orange flavour, just sampled eucalyptus and oregano-flavoured diet and rejected mint flavour [29]. These findings contrast with our findings where orange was the least preferred flavour among the liquid aroma flavours, and juicy orange flavour did not differ in preference among the powder-aroma flavours offered. Lambs preferred milky over fenugreek in this study, in agreement with the findings of Nedelkov et al. [30], but the preference for anise over fenugreek was contrary to the findings of Harper et al. [11], where cows were found to prefer fenugreek over anise. The discrepancies between studies may be due to the animal species used or the flavour inclusion rate. It was surprising to see a preference for capsicum over nexulin, since the active ingredient in both products is capsaicin, the only difference being that nexulin is rumen protected while capsicum is not. The preference for capsicum could be due to fat granules included in the capsicum to reduce its pungency [31]. The lack of preference for nexulin was in agreement with the findings of Oh et al. [32], who reported a sharp decrease in DMI after supplementing cows’ total mixed ration with nexulin. In this study, consumption of the flavoured grain mixture did not increase relative to control, in agreement with studies carried out in cows [11] and lambs [30]. Calves with good appetite (above 600 g/d) also showed a lack of preference when fed an orange or unflavoured starter ration, while those with a low appetite (below 600 g/d) preferred an orange flavoured ration. Thus, it could be concluded that flavour preference is dependent on appetite. Therefore, the lack of preference relative to control in this study may be due to the high feed intake level [10], since daily feed intake was high at 69.3–71.2 g DM/Kg BW^0.75^.

In our study, the flavours were mixed with a corn and barley grain mixture, while the unflavoured grain mixture acted as the control. The grain mixture might have stimulated intake similar to sucram, resulting in the lack of significant difference. A study conducted to examine the effects of sucram on the dietary preference of feedlot cattle reported that calves fed a 65% concentrate diet ad libitum supplemented with sucram at 200 mg/kg did not show an increase in feed intake after sucram supplementation. The authors stated that sweetened feeds might have a similar effect to high-concentrate diets. Hence a high-concentrate diet might stimulate the feed intake as sweeteners resulting in similar feed intake between animals fed a high-concentrate diet with or without sucram supplement [33].

### 4.2. DMI, BW Gain, Nutrient Apparent Digestibility and Energy Balance

#### 4.2.1. DMI and BW Gain

Capsicum did not increase the feed intake compared to the control group in this study, similar to previous studies that reported no difference in dairy cows [16,31,34,35]. However, capsicum was reported to increase DMI in beef cattle [36,37]. The inconsistent results may be due to differences in the inclusion rate of capsicum in the diets [31]. In the current study, the capsicum supplemented diet was observed to increase ADG. In agreement with our results, although in different species, capsaicin supplementation at 150 ppm increased body weight gain in Pekin ducks [38]. El-Tazi [39] also reported that 0.5%, 0.75% and 1% capsaicin inclusion in broiler chicks’ diet improved body weight gain and feed conversion ratio. This could be due to the antimicrobial properties of capsaicin that alleviate harmful microbes in the gastrointestinal tract to improve epithelial integrity, such as an increase in the mucosa and sub-mucosa thickness of the small intestine and absorption surface [40,41].

Sucram did not increase body weight gain and feed intake when supplemented alone or combined with capsicum in this study. Discrepancies exist in the available literature on the effect of artificial sweeteners supplemented alone or in combination with other flavours. The inclusion of natural or artificial high-intensity sweeteners was reported to increase feed intake and body weight gain in weanling pigs [42] and horses [43]. However, some studies failed to demonstrate a positive effect of sweeteners on feed intake and growth performance in pigs [44,45,46]. In addition, sucram was reported to have no effect on DMI in feedlot steers [33] or on the total BW gain and ADG in stressed calves [47]. Stevia [46] and a combination of saccharin, neohesperidin and dihydrochalcone [42] were also reported to have no effect on growth performance in pigs, in agreement with our results. Since sweeteners and high-concentrate diets may have a similar effect on feed preference [33], the discrepancy in the effect of sweeteners on feed intake might be related to differences in diet composition. In our study, lambs were offered high-energy diets.

The higher weight gain in the males compared to the female lambs was in congruence with previous studies reported for the Blackhead sheep [48], Assaf lambs [49] and Morada Nova lambs [50]. The higher weight gain in the male lambs may be due to the higher feed intake observed. Our findings of higher voluntary feed intake in the male than in the female lambs concur with the findings of De Araújo et al. [50], but contrast with the findings of Rodríguez et al. [49] and Aregheore [48] who reported no effect of sex on feed intake in sheep. The difference in feed intake may be due to the difference in body weight. Generally, males have a larger body size than female sheep. A sexual size dimorphism study reported a 1.41 ratio of males to females in domesticated sheep [51]. Ruminant livestock with higher body weight consume more feed compared to their counterparts with lower body weight [52]. Hence, the higher feed intake observed in male than the female lambs was expected. The effect of sex on feed intake may also be due to differences in leptin levels between males and females [53]. However, leptin was not measured in this study.

#### 4.2.2. Nutrient Apparent Digestibility

In ruminants, nutrient digestibility is influenced by the degradation of feed ingredients by ruminal microbes, enzymatic digestion in the lower gut and passage rate [54,55]. Although sweeteners are reported to interfere with gut microbe numbers and balance [14,15] which may affect essential gut functions, such as nutrient metabolism, immune system functioning and inhibition of pathogens [14], no effect of sucram on feed digestibility was observed in this study. Our findings are in agreement with the study of Ponce et al. [47] who reported no effect of sucram on the apparent total tract digestibility of DM, OM, CP or NDF in calves. The lack of difference may be due to the similar feed intake that may have resulted in a similar passage rate of the digester through the gastrointestinal tract [56].

Capsicum is reported to have no effect on volatile fatty acids and ammonia production (concentration) in the rumen [31,35]. Likewise, purine derivative excretion, as an indicator of ruminal microbial synthesis, was not affected by capsicum supplementation in dairy cows [16]. Although major feed digestion in ruminants takes place in the rumen, the lower gut contribution cannot be overlooked as 35.3%, 21.2% and 19.5% of OM, cell wall and starch total tract digestibility, respectively, are reported to take place in the lower gastrointestinal tract [57]. In the current study, capsicum tended to increase the EE digestibility. This increase may be due to the stimulatory effect of capsaicin on the digestive enzyme secretion in the lower gut. Capsaicin was reported to increase lipase secretion in rats [58]. Oh et al. [16] reported that capsaicin supplementation in dairy cows increased the apparent total-tract digestibility of DM, OM and CP, but Ali et al. [38] found no effect of capsaicin supplementation on CP digestibility. The discrepancies in the results may be due to differences in the capsaicin dosage or the formulation used [31]. For instance, the study of Oh et al. [16] used rumen-protected Capsicum oleoresin while capsicum used in our study was not rumen-protected.

#### 4.2.3. Energy Balance

In ruminants, dietary carbohydrates are fermented by rumen microflora into short-chain fatty acids [59]. However, up to 50% of starch escapes undegraded into the small intestine, especially when the starch source is corn, sorghum or legumes, where it undergoes enzymatic degradation into monosaccharides [55,59]. Apical monosaccharide absorption takes place through the intestinal brush border membrane and requires *SGLT1* for glucose, galactose, the facilitated transporters glucose transporter 2 (*GLUT2*) and glucose transporter 5 (*GLUT5*) for fructose [60]. Monosaccharides are sweet tasting, hence they are sensed by the intestinal *T1R2*+*T1R3* heterodimer to induce glucose absorption in most mammalian species via the expression and activity of *SGLT1* [7,61]. Sweeteners in low concentrations are reported to be more effective activators of *T1R2* and *T1R3* [7] and, consequently, higher glucose absorption [13]. In the current study, sucram had no effect on energy intake or retention, as shown in GEI, DE, FE, GasE, ME, HiE, NE and UE. Since *SGLT1* expression is influenced by diet composition [62], the high-energy content in the basal diet in the current study may have saturated the *SGLT1* mechanism, therefore, masking the effect of sucram.

Capsicum did not influence the energy balance of lambs in this study. Although capsicum has been reported to increase the lipases and disaccharidases in the intestines of rats [58], and tends to increase feed efficiency in dairy cows [23], there is a paucity of the peer-reviewed literature on the effect of capsicum on energy balance in ruminants. However, consumption of capsaicin is reported to promote a negative energy balance in human studies [63]. Capsaicin is reported to influence gut microbiota in vitro [64], and previous studies stated that alterations of specific bacteria influence the regulation of glucose homeostasis. For example, an increased abundance of *Roseburia* is positively correlated with glucose homeostasis [65] through increased secretion of glucagon-like peptide 1 (GLP-1), a hormone that increases glucose absorption by increasing the expression of *SGLT1* [66]. In addition, activation of the capsaicin receptor transient receptor potential cation channel subfamily V member 1 (TRPV1) is reported to enhance GLP-1 secretion [67]. Therefore, there is need for more studies to evaluate the effect of capsaicin on energy balance and the underlying mechanism in ruminants.

### 4.3. T1R2, T1R3 and SGLT1 Relative Gene Expression

Sweet taste receptors sense luminal glucose to initiate a cyclic adenosine monophosphate-protein kinase A (cAMP–PKA) pathway that enhances *SGLT1* expression in sheep [17]. Sweeteners, such as sucram, are considered more effective activators of *T1R2* and *T1R3* because they are 600 times “sweeter” than glucose [7]. In the current study, sucram acted synergistically with capsicum to increase *T1R3* gene expression. Sucralose was reported to act synergistically with glucose to activate *T1R2*+*T1R3* heterodimers to increase glucose absorption through translocation of *GLUT2* into the apical membrane of the enterocyte via a phospholipase C (PLC) βII-dependent pathway in rats [68].

Sucram alone had no effect on *T1R2* and *T1R3* gene expression at the proximal jejunum in this study. A lack of effect on sweet taste receptors may be due to the existence of other sweet sensors, saturation or overload of substrates. A study by Damak et al. [69] reported that *Tas1r3* knockout mice preferred high concentrations of sucrose and glucose. The authors attributed the findings to the presence of other less sensitive sugar receptors, such as the dpa locus, influencing neural and behavioural sensitivities to sucrose in mice [70]. Furthermore, Kusuhara et al. [71] reported that *T1R1*−/− mice showed low chorda tympani nerve responses to sweeteners, and had a lower proportion of sweet responsive cells compared to *T1R1*+/− mice. These results show that *T1R1*-expressing cells partly contribute to a sweet sensitivity. Hence, the *T1R2* and *T1R3* expression measured in the current study might not be the only representative of the overall arsenal responsive to sweet substrates.

The *SGLT1* gene expression is influenced by sweet taste receptor signalling [7]. Supplementing adult cows diet containing an 80:20 ratio of ryegrass hay-to-concentrates feed with sucram caused an over 7-fold increase in *SGLT1* protein abundance associated with increased mRNA expression [7]. On the contrary, rats showed no receptor/sensor-mediated change in the level of apical *SGLT1* after 30 min of glucose and sucralose perfusion in the jejunum [68]. Moreover, sucram-treated calves had similar *SGLT1* mRNA abundance as the control calves maintained on a starter concentrate diet [7]. Since monosaccharides are sweet tasting [7], dietary glucose might have saturated sweet taste receptors causing a lack of effect after sucram supplementation observed in this study. In addition, *SGLT1* regulation may occur during translation or post-translation [72]. The *SGLT1* expression and activity exhibits a diurnal rhythm [73], and glucose transporters are expressed at different magnitudes along the gastrointestinal tract [74], hence the difference in findings of the different studies.

Direct effects of capsaicin on taste receptor cells are not well understood. Immunohistochemical analysis showed that TRPV1, a receptor activated by capsaicin [75], is localised in the taste cells of circumvallate papillae in rats [19]. Capsaicin also inhibits the potassium currents of taste receptor cells isolated from the rats’ circumvallate papillae [76]. These studies suggest that capsaicin enhances or modifies the sweet and bitter taste perception in rats’ circumvallate papillae in vivo [19]. Gu et al. [20] reported that repeated oral exposure to capsaicin increased the consumption of sucrose and saccharin-sweetened solutions in rats, although it decreased the mRNA expression of sweet taste receptors in the circumvallate papillae. In this study, capsicum alone had no significant effect on the *T1R2*, *T1R3* and *SGLT1* gene expression, but supplementation with a mix of capsicum and sucram increased the expression of T1R3. The increase in *T1R3* gene expression may indicate that capsaicin and sucram caused an interaction in the taste transduction pathway that gave an impression of high sweet tasting stimuli in the small intestine’s lumen. The *T1R3* is reported to form a *T1R3*/*T1R3* homodimer at high monosaccharide and disaccharide concentrations to increase uptake [77]. However, the increase in *T1R3* gene expression did not amount to an increase in the *SGLT1* expression, which may explain the lack of difference in energy balance. Furthermore, the mRNA abundance does not accurately predict the protein abundance in some cases [21], hence the increase in the *T1R3* gene expression may have not increased the *T1R3* protein abundance. Therefore, there is need for more studies to evaluate the underlying mechanism behind the effect of the capsicum and sucram mixed flavours on sweet taste receptors.

## 5. Conclusions

Overall, we found that flavours had a significant effect on preference. Capsicum increased weight gain and tended to increase either the extract digestibility with no clear effect on energy balance or expression of the evaluated genes. On the contrary, supplementing lambs with sucram combined with capsicum led to higher *T1R3* gene expression with no improvement in feed intake, digestibility or weight gain. Our results demonstrate that supplementing lambs with capsicum together with sucram has a synergistic effect that increases the sweet taste receptors gene expression. Therefore, our hypothesis that ruminants prefer some flavours over others and supplementing feed with their preferred flavours after weaning increases feed acceptance, intestinal expression of sweet taste receptors and monosaccharide transporters leading to increased feed efficiency was partially accepted. Our study demonstrates that flavours can be employed as a less costly method to increase weight gain in young ruminants. However, further studies are required to examine the metabolic pathway in which capsicum increases weight gain and determine a standard flavour inclusion rate in the diets of different farm animals.

## Figures and Tables

**Figure 1 animals-13-01417-f001:**
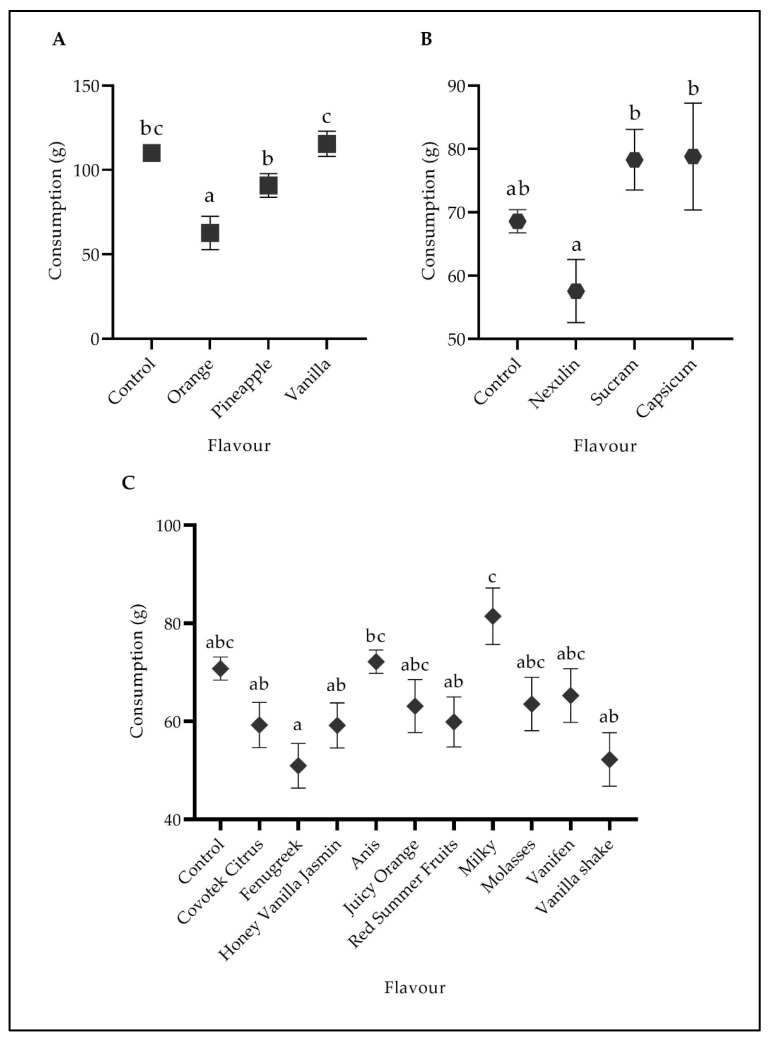
Consumption of the grain mixture flavoured with liquid aroma ((**A**); *p* < 0.001), non-aroma ((**B**); *p* = 0.020) or powder-aroma flavours ((**C**); *p* < 0.001) by sheep offered free choice for 5 min. Means ± standard error with different letters (a–c) differ significantly (*p* ≤ 0.05).

**Figure 2 animals-13-01417-f002:**
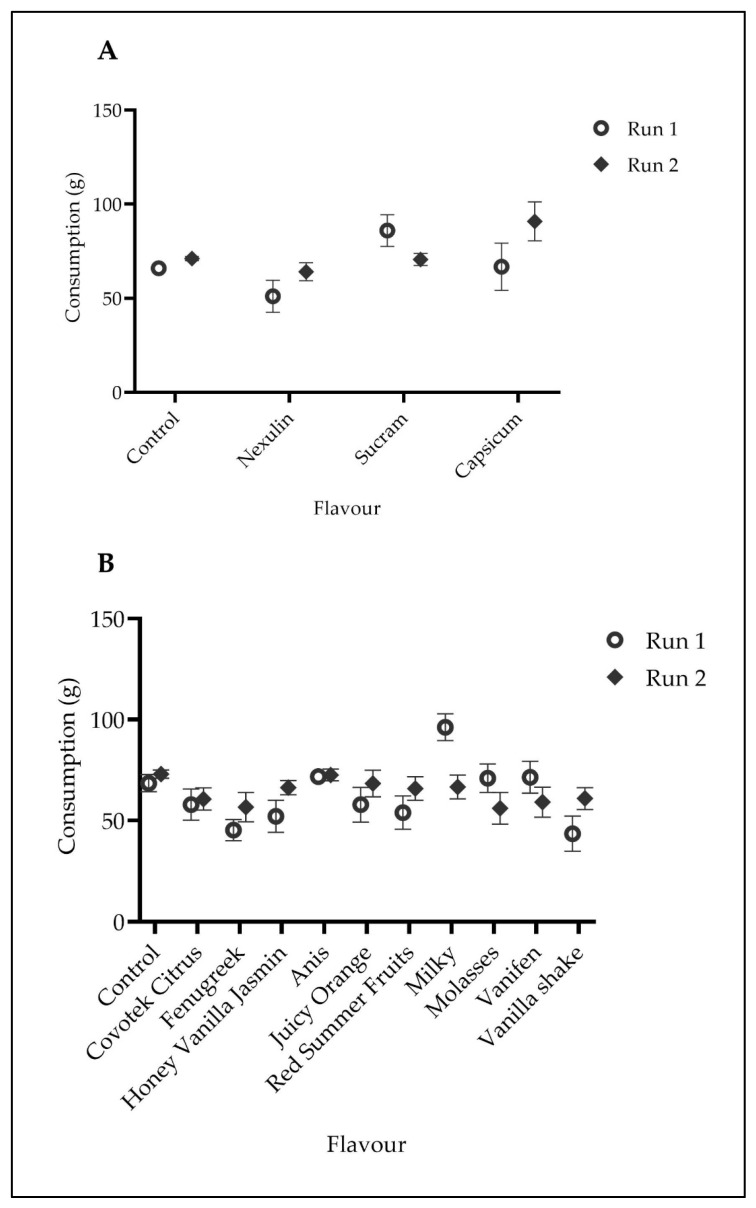
Consumption of the grain mixture flavoured with liquid non-aroma ((**A**); *p* = 0.023) or powder-aroma flavours ((**B**); *p* < 0.001) during the first (Run 1) or subsequent (Run 2) exposure presented as the mean ± standard error.

**Figure 3 animals-13-01417-f003:**
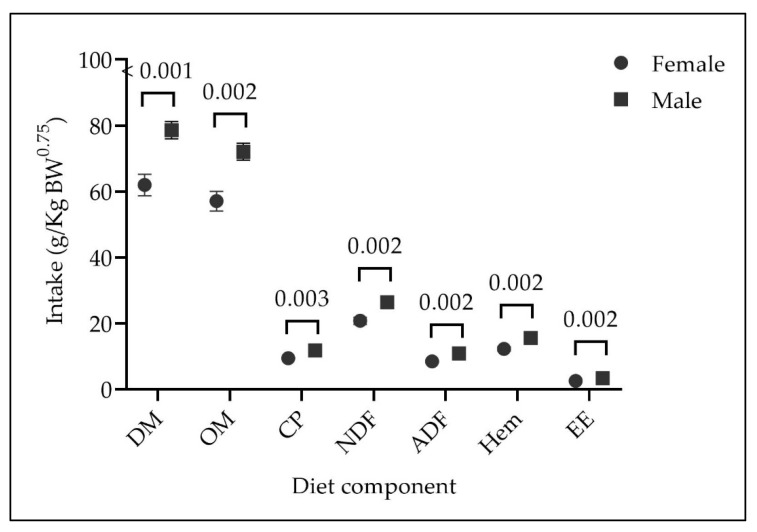
Difference in the daily dry matter (DM), organic matter (OM), crude protein (CP), neutral detergent fibre (NDF), acid detergent fibre (ADF), hemicellulose (Hem) and ether extract (EE) intake between females and males fed unflavoured diet (control) or diet flavoured with sucram, capsicum or both expressed as the mean ± standard error.

**Figure 4 animals-13-01417-f004:**
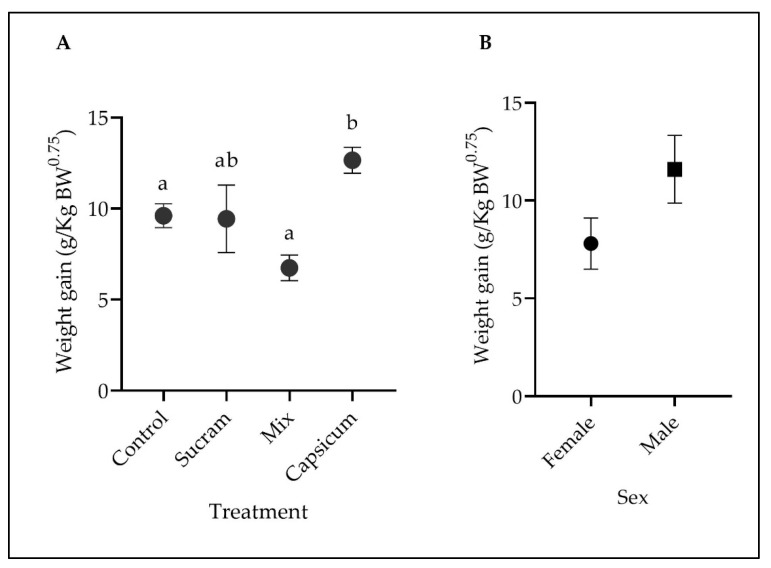
Effect of flavour ((**A**); *p* = 0.049) and sex ((**B**); *p* = 0.024) on daily weight gain per metabolic body weight of sheep fed unflavoured diet (control) or diet flavoured with sucram, capsicum or both (mix) expressed as the mean ± standard error. Means ± standard error with different letters (a, b) differ significantly (*p* ≤ 0.05).

**Figure 5 animals-13-01417-f005:**
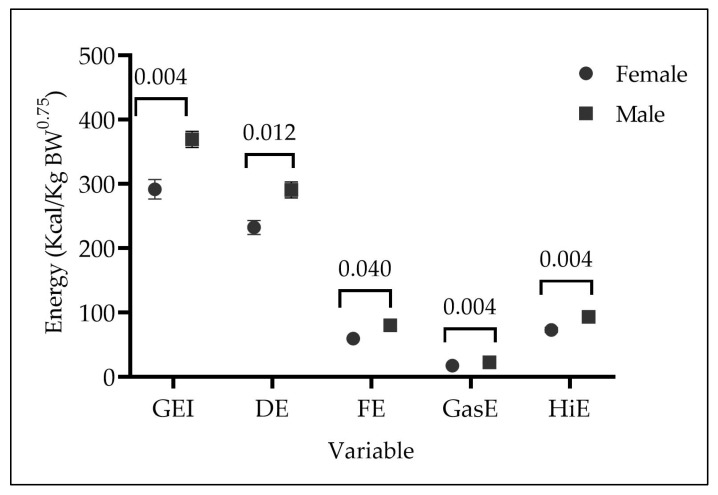
Effect of sex on GEI (gross energy intake), DE (digestible energy), FE (faecal energy), GasE (combustible gas energy) and HiE (heat increment energy) of females and males fed unflavoured diet or diet flavoured with sucram, capsicum or both. Data are expressed as the mean ± standard error.

**Figure 6 animals-13-01417-f006:**
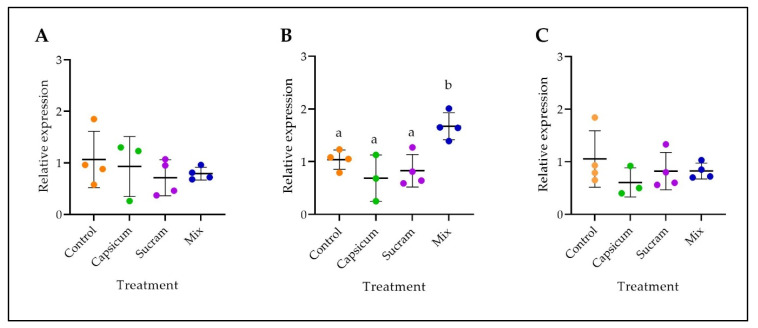
Relative gene expression of the taste receptor family 1 member 2 ((**A**); T1R2; *p* = 0.193), taste receptor family 1 member 3 ((**B**); T1R3; *p* = 0.005) and sodium-dependent glucose transporter isoform 1 ((**C**); SGLT1; *p* = 0.378) in the proximal jejunum of sheep fed an unflavoured diet (control) or diet flavoured with sucram, capsicum or both (mix). Means ± standard error with different letters (a, b) differ significantly (*p* ≤ 0.05).

**Table 1 animals-13-01417-t001:** Composition of the total mixed ration (TMR) and pellets (expressed as % DM unless otherwise stated).

Feeds	TMR	Pellets
Corn grains	13.9	10.4
Wheat bran	19.6	1.7
Gluten feed		5.3
Barley grain		11.4
Soybean meal		31.2
Sunflower meal	9.5	6.1
Wheat grain		11.3
Dried distiller grains		7.0
Vegetable oil		1.2
Grass hay	15.1	
Wheat silage	11.3	
Wheat straw	15.8	
Citrus pulps	2.5	
Soybean hulls	9.5	0.9
Limestone	1.3	5.0
Salt	0.5	5.0
NH_4_Cl		0.5
Na bicarbonate		1.0
Cu sulphate	0.02	
Vitamin mix *	0.9	1.9
Chemical composition		
Moisture,	37.1	10.4
Crude protein	12.5	26.0
Crude fat	2.76	3.63
ME, Mcal/kg DM	2.30	2.70
NDF	48.2	16.6
Ca	0.85	2.30
P	0.46	0.48
Cu	0.49	0.51
Vit A, IU	5862	14,000

* Vitamin premix 3678^®^, Bar Magen LTD, Israel: 8MIU Vit A, 1.6MIU Vit E, 20KIU Vit E, 15 g Anilox, 30 g Mn, 100 g Zn, 20 g Fe, 0.5 g I, 1 g Co, 0.1 g Se, 3 kg NaCl, 3 kg Na_2_SO_4_, 5 kg NH_4_Cl, 0.5 kg limestone.

**Table 2 animals-13-01417-t002:** Primers used for the real-time PCR analysis of gene expression in the proximal jejunum.

Gene ^1^	Primer F (5′-3′)	Primer R (5′-3′)	Product Size	Gene Accession Number
*G6PDH*	ATTGTGGAGAAGCCCTTCGG	GGTAGTGGTCGATGCGGTAG	106	NM_001093780.1
*B2M*	TGCTGAAGAACGGGGAGAAG	GAACTCAGCGTGGGACAGAA	92	NM_001009284.2
*GAPDH*	GGCGTGAACCACGAGAAGTA	GGCGTGGACAGTGGTCATAA	141	NM_001190390.1
*T1R2*	TTGGCCCCAAGTGTTACCTG	CCCTGGATCACGCTGTTGAA	76	KC469057.1
*T1R3*	TGACCGATGGGCTGCTATAC	GCAGAGGTGAAGTGCGTGG	80	XM_015099284.1
*SLC5A1* (*SGLT1*)	GAGGGTACAGTGCCTTCGTG	GGATCGCGGAAGATGTGGAA	127	NM_001009404.1

^1^ *G6PDH* = glucose-6-phosphate dehydrogenase; *B2M* = beta-2 microglobulin; *GAPDH* = glyceraldehyde 3-phosphate dehydrogenase; *T1R2* = taste receptor family 1 member 2; *T1R3* = taste receptor family 1 member 3; *SLC5A1* = solute carrier family 5 member 1 and *SGLT1* = sodium-dependent glucose transporter isoform 1.

**Table 3 animals-13-01417-t003:** Effect of the flavours on daily feed intake expressed in grams per metabolic body weight (g/Kg BW^0.75^) in lambs.

	Treatments ^1^					
Variable ^2^	Control	Sucram	Mix	Caps	SEM ^3^	*p*-Value
DM	69.3	71.0	69.7	71.2	7.33	0.934
OM	63.8	64.7	64.3	65.5	6.74	0.964
CP	10.5	10.7	10.6	10.8	10.64	0.973
NDF	23.3	23.6	23.5	24.0	2.30	0.955
ADF	9.5	9.7	9.7	9.8	0.93	0.952
Hemicellulose	13.8	13.9	13.9	14.2	1.38	0.953
EE	3.0	3.0	3.0	3.1	0.35	0.893

^1^ Control = basal diet without flavour; sucram = basal diet supplemented with 150 g/ton sucram; caps = basal diet supplemented with 150 g/ton capsicum and mix = basal diet supplemented with 150 g/ton sucram and capsicum at 1:1 ratio; ^2^ DM = dry matter; OM = organic matter; CP = crude protein; NDF = neutral detergent fibre; ADF = acid detergent fibre and EE = ether extract; ^3^ standard error of the mean.

**Table 4 animals-13-01417-t004:** Effect of flavours on the total tract feed digestibility in lambs (expressed in percentage).

	Treatment ^1^					
Variable ^2^	Control	Sucram	Mix	Caps	SEM ^3^	*p*-Value
DM	73.6	73.9	74.0	77.4	2.53	0.469
OM	77.3	77.3	77.7	80.4	2.21	0.476
CP	73.1	73.8	73.6	77.7	2.11	0.395
NDF	57.4	55.2	55.7	62.0	4.06	0.349
ADF	45.9	43.9	44.4	51.2	5.34	0.493
Hemicellulose	65.3	63.1	63.5	69.6	3.45	0.265
EE	73.3	65.8	68.4	76.6	4.43	0.083

^1^ Control = basal diet without flavour; sucram = basal diet supplemented with 150 g/ton sucram; caps = basal diet supplemented with 150 g/ton capsicum and mix = basal diet supplemented with 150 g/ton sucram and capsicum at 1:1 ratio; ^2^ DM = dry matter; OM = organic matter; CP = crude protein; NDF = neutral detergent fibre; ADF = acid detergent fibre and EE = ether extract; ^3^ standard error of the mean.

**Table 5 animals-13-01417-t005:** Effect of the flavours on the energy balance expressed in kilocalories per metabolic body weight per day (Kcal/day/Kg BW^0.75^).

	Treatment ^1^					
Variable ^2^	Control	Sucram	Mix	Caps	SEM ^3^	*p*-Value
GEI	326.8	331.7	329.0	333.7	35.28	0.983
DE	255.9	260.2	256.1	273.6	31.05	0.834
FE	70.9	71.5	73.0	63.1	10.70	0.756
UE	6.51	9.94	6.42	7.57	3.55	0.723
GasE	19.6	19.9	19.7	19.9	0.87	0.984
ME	217.9	211.5	191.3	207.6	9.71	0.267
HiE	81.7	82.9	82.3	82.8	3.63	0.983
NE	141.2	136.7	124.0	135.0	5.97	0.254

^1^ Control = basal diet without flavour; sucram = basal diet supplemented with 150 g/ton sucram; caps = basal diet supplemented with 150 g/ton capsicum and mix = basal diet supplemented with 150 g/ton sucram and capsicum at 1:1 ratio.; ^2^ GEI = gross energy intake; DE = digestible energy; FE = faecal energy; UE = urine energy; GasE = combustible gas energy; ME = metabolizable energy; HiE = heat increment energy; NE = net energy. ^3^ Standard error of the mean.

## Data Availability

Data available from the corresponding author.

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
