# Peer review of "Diet Preference, Feed Efficiency and Expression of the Sodium-Dependent Glucose Transporter Isoform 1 and Sweet Taste Receptors in the Jejunum of Lambs Supplemented with Different Flavours"

_animals, 2023, doi:10.3390/ani13081417_

Round 1
Reviewer 1 Report
Subject of the study falls within journal’ objectives and provide an alternative to improve feed acceptance in lambs. In order to accept the manuscript, some concerns are discussed:
The abstract should be re-written in no more than 200 words. I suggest to revisited the introduction and provide more evidence to justify the study (importance of the expression of evaluated genes might be mention). Materials and Methods and discussion are hard to follow. I highly recommend including more recent references, due to more than 50% of the references are more than 10 years old since they were published; also highlight the most relevant ones, the study has too many references. Author must include the DOI for all references.
Throughout all the manuscript, English writing needs to be checked.
Introduction section
Line 59-61: The author mentions “animals” but the one cited study only refers to pigs. “Pluske, J.R.; Hampson, D.J.; Williams, I.H. Factors influencing the structure and function of the small intestine in the weaned 545 pig: a review. Livest. Prod. Sci. 1997, 51, 215–236”
Line 77-83: Better not to mention “young ruminants”, the study focuses only on lambs.
Materials and methods section.
In the section 2.1 Preference Study: please provide a reference for the protocol used. Also, I recommend specifying if the flavor additives are non-nutritive or are energetic compounds.
Section 2.2. The title of section 2.2 should include sampling or biopsy of intestinal tissue.
On two occasions (lines 128 and 136) it is mentioned that a crossover experiment was carried out with a 4 x 2 arrangement. Write in a simple way without duplicating information. Please, provide a reference for the intestine tissue collection protocol.
Section 2.4
Line 193 and 195: “NanoDrop ND-1000 spectrophotometer” and “qPCRBIO cDNA synthesis kit” please provide the manufacturer information.
Section 2.5
Statistical analysis: Is the initial BW of each period considered as a covariate in the analysis?. It is not mentioned how the effect of baseline weight was controlled for in the study.
Line 227: “The model included flavour, run (first or subsequent exposure to the flavour), 227 sex, flavour-by-run interaction and flavour-by-sex interaction as the fixed effects”. No random effect was considered in the model?
Results Section
Line 241: Abovementioned, significant differences were considered p<0.05. Please correct “p>0.1” to “p>0.05”.
I recommend that the results in Figures 1-5 be shown as the mean ± standard error, rather than the standard deviation. This will help to better visualize the significance.
Discussion section
Line 370-372: I doubt the cited reference support the idea of this sentence (Reference 33).
Line 419-420: Non ruminal fermentation characteristics were evaluated in this study, better not to discuss/concluded this way.
Section 4.3 T1R2, T1R3 and SGLT1 Relative Gene Expression: I suggest discussing the metabolic implications of the increase found in the T1R3 gene expression, the author may provide a potential explanation for this improvement.
Line 513-514: “This may increase glucose absorption in the intestine and thus increase metabolic energy efficiency for growth”. Better not to mention this idea in the conclusion (suits better in the discussion section).
Author Response
We thank the reviewer for providing feedback and suggestions that will help improve the quality of this manuscript.
Response to Reviewers comments and suggestions
Reviewer 1
Subject of the study falls within journal’ objectives and provide an alternative to improve feed acceptance in lambs. In order to accept the manuscript, some concerns are discussed:
Response: We thank the reviewer for providing feedback and suggestions that will help improve the quality of this manuscript. We have addressed all the concerns individually as described below.
The abstract should be re-written in no more than 200 words.
Response: The abstract has been re-written in 200 word, as presented below.
“This study investigated the effect of dietary flavour supplements on preference, feed efficiency and expression of the sweet taste receptor family 1 members 2 and 3 (T1R2+T1R3), and sodium-glucose linked transporter 1 (SGLT1) genes in the lambs small intestines. Eight, five-months-old, Israeli crossbred Assaf lambs were offered 16 different non-nutritive commercial flavours in rolled barley and ground corn. Capsicum and sucram were the most preferred non-aroma flavours (p = 0.020), while milky (p < 0.001) was the most preferred powder-aroma flavour. For the metabolic and relative gene expression study, eight lambs were randomly assigned to either sucram, capsicum, mix containing sucram and capsicum at 1:1 ratio or no flavour for control in a 4x2 cross-over design. Total collection of urine (females only), feces and refusals was carried out and T1R2, T1R3 and SGLT1 relative gene expression evaluated from the proximal jejunum biopsies. Flavour had no significant effect on feed intake (p = 0.934), but capsicum increased average daily weight gain per metabolic body weight (p = 0.049). The T1R3 gene was expressed highest in the mix treatment (1.7; p = 0.005). Collectively, our findings indicate that flavours can be used to motivate feed acceptance and improve weight gain in lambs.”
I suggest to revisited the introduction and provide more evidence to justify the study (importance of the expression of evaluated genes might be mention).
Response: We have revisited the introduction and added a paragraph to justify importance of the expression of the evaluated genes.
‘Furthermore, inclusion of sweeteners or capsaicin-based flavours in diets has been re-ported to influence feed efficiency and nutrients metabolism [13–16]. Dietary sweeteners bind to the sweet taste receptors and induce an intracellular transduction cascade that activates the sodium-glucose linked transporter 1 (SGLT1), and subsequently, glucose uptake [17,18]. On the other hand, oral exposure to capsaicin was reported to increase consumption of sucrose and saccharin-sweetened solutions in rats [19] and apparent total-tract feed digestibility in dairy cows [16], but the underlying mechanism is not well under-stood. Since gene expression may indicate protein abundance [20], there is need for studies evaluating the expression of the sweet taste receptors and glucose transporters to un-ravel the underlying mechanism of flavours on feed efficiency..’
Materials and Methods and discussion are hard to follow.
Response: The Materials and Methods section has been edited for clarity. “….Lambs were allowed seven days adaptation period to the basal diet and experimental facility. The preference test was then conducted according to the modified protocol of Harper et al. [11] using 16 different non-nutritive commercial flavours: …… An unflavoured grains mixture was used as the control, offered at a random time similar to the flavoured treatments. The treatments were replicated twice, once in two days, to prevent carry-over effects and all the lambs had access to all the flavours……The lambs (one male and one female) were assigned to either sucram, capsicum, mix (su-cram and capsicum at 1:1 ratio) or no flavour control supplemented in the feed. The feed consisted of total mixed ration, whole corn grain and pellets in 50:25:25 ratio (weight as is), respectively, formulated to ensure 15.7% CP and 2.68 Mcal/kg metabolisable energy (ME) on DM basis (Table 1). Every morning, feed and flavours were weighed and mixed thoroughly by hand for two minutes, partitioned into 12 equal meal portions and loaded into the automatic feeders. Total collected urine and feces were weighed and stored at −20 °C until chemical analysis. The treatments were switched at the end of the first period and the experiment repeated for the second period. Sheep were weighed at the beginning and at the end of each period of the experiment to determine weight gain (Scales Galore NY, U.S.A) and daily weight gain expressed in g/Kg BW0.75 to control for baseline weight variability between periods.”
As mentioned by Reviewer 2, we tried to present the discussion as adequately as possible. However, we have deleted Lines 419-422 “A previous study reported an improvement of feed digestibility by capsaicin supplementation in avian species [31], and the authors attributed the improvement to the activation of the sympathetic nervous system, increased secretion of digestive enzymes and bile acids [53].” and added eight lines in section 4.3 “The increase in T1R3 gene expression may indicate that capsaicin and sucram caused an interaction in the taste transduction pathway that gave an impression of high sweet tasting stimuli in the small intestines lumen. The T1R3 is reported to form a T1R3/T1R3 homodimer at high monosaccharides and disaccharides concentration to increase uptake [77]. However, the increase in T1R3 gene expression did not amount to an increase in the SGLT1 expression, which may explain the lack of difference in energy balance. Also, the mRNA abundance does not accurately predict the protein abundance in some cases [21], hence the increase in the T1R3 gene expression may have not increased the T1R3 protein abundance.”
I highly recommend including more recent references, due to more than 50% of the references are more than 10 years old since they were published; also highlight the most relevant ones, the study has too many references.
Response: The reviewer’s recommendation was taken on board. Thirteen articles were deleted to reduce the number of reference or replaced with more recent references, as indicated below.
Deleted references
Goatcher, W.D.; Church, D.C. Review of some nutritional aspects of the sense of taste. J. Anim. Sci. 1970, 31, 973–981.
Pluske, J.R.; Hampson, D.J.; Williams, I.H. Factors influencing the structure and function of the small intestine in the weaned pig: a review. Livest. Prod. Sci. 1997, 51, 215–236
McLaughlin, C.L.; Baile, C.A.; Buckholtz, L.L.; Freeman, S.K. Preferred flavors and performance of weanling pigs. J. Anim. Sci. 1983, 56, 1287
Rolls, B.J. How variety and palatability can stimulate appetite. Nutr. Bull. 1979, 5, 78–86.
Treit, D.; Spetch, M.L.; Deutsch, J.A. Variety in the flavor of food enhances eating in the rat: A controlled demonstration. Physiol. Behav. 1983, 30, 207–211.
Dyer, J.; Hosie, K.B.; Shirazi-Beechey, S.P. Nutrient regulation of human intestinal sugar transporter (SGLT1) expression. Gut 1997, 41, 56–59.
Shirazi-Beechey, S.P.; Gribble, S.M.; Wood, I.S.; Tarpey, P.S.; Beechey, R.B.; Dyer, J.; Scott, D.; Barker, P.J. Dietary regulation of the intestinal sodium-dependent glucose cotransporter (SGLT1). Biochem. Soc. Trans. 1994, 22, 655–658.
Walters, W.A.; Xu, Z.; Knight, R. Meta-analyses of human gut microbes associated with obesity and IBD. FEBS Lett. 2014, 588, 4223–4233.
Johnston, M. Feasting, fasting and fermenting: glucose sensing in yeast and other cells. Trends Genet. 1999, 15, 29–33
Margolskee, R.F.; Dyer, J.; Kokrashvili, Z.; Salmon, K.S.H.; Ilegems, E.; Daly, K.; Maillet, E.L.; Ninomiya, Y.; Mosinger, B.; Shirazi-Beechey, S.P. T1R3 and gustducin in gut sense sugars to regulate expression of Na+-glucose cotransporter 1. Proc. Natl. Acad. Sci. U. S. A. 2007, 104, 15075–80
Corpe, C.P.; Burant, C.F. Hexose transporter expression in rat small intestine: effect of diet on diurnal variations. Am. J. Physiol. - Gastrointest. Liver Physiol. 1996, 271, G211–G126.
Alderman, G.; Cottrill, B.R. Energy and protein requirements of ruminants: an advisory manual prepared by the AFRC Technical Committee on Responses to Nutrients; Alderman, G., Cottrill, B.R., Eds.; CAB International: Wallingford, UK, 1993; ISBN 9780851988511.
Rolland, F.; Winderickx, J.; Thevelein, J.M. Glucose-sensing mechanisms in eukaryotic cells. Trends Biochem. Sci. 2001, 26, 310–317.
Added references
Favreau-Peigné, A.; Baumont, R.; Ginane, C. Food sensory characteristics: Their unconsidered roles in the feeding behaviour of domestic ruminants. Animal 2013, 7, 806–813.
De Souza Teixeira, O.; Kuczynski da Rocha, M.; Mendes Paizano Alforma, A.; Silva Fernandes, V.; de Oliveira Feijó, J.; Nunes Corrêa, M.; Andrighetto Canozzi, M.E.; McManus, C.; Jardim Barcellos, J.O. Behavioural and physiological responses of male and female beef cattle to weaning at 30, 75 or 180 days of age. Appl. Anim. Behav. Sci. 2021, 240, 105339.
Abdelsattar, M.M.; Vargas-Bello-Pérez, E.; Zhuang, Y.; Fu, Y.; Zhang, N. Impact of dietary supplementation of β-hydroxybutyric acid on performance, nutrient digestibility, organ development and serum stress indicators in early-weaned goat kids. Anim. Nutr. 2022, 9, 16–22.
McCoard, S.A.; Cristobal-Carballo, O.; Knol, F.W.; Heiser, A.; Khan, M.A.; Hennes, N.; Johnstone, P.; Lewis, S.; Stevens, D.R. Impact of early weaning on small intestine, metabolic, immune and endocrine system development, growth and body composition in artificially reared lambs. J. Anim. Sci. 2020, 98, 1–11
Bhutta, H.Y.; Deelman, T.E.; Ashley, S.W.; Rhoads, D.B.; Tavakkoli, A. Disrupted circadian rhythmicity of the intestinal glucose transporter SGLT1 in Zucker diabetic fatty rats. Dig. Dis. Sci. 2013, 58, 1537–1545.
Diaz-sanchez, S.; Souza, D.D.; Biswas, D.; Hanning, I. Botanical alternatives to antibiotics for use in organic poultry production. Poult. Sci. 2015, 94, 1419–1430.
Fattori, V.; Hohmann, M.; Rossaneis, A.; Pinho-Ribeiro, F.; Verri, W. Capsaicin: Current understanding of its mechanisms and therapy of pain and other pre-clinical and clinical uses. Molecules 2016, 21, 844.
Neumann, N.J.; Fasshauer, M. Added flavors: potential contributors to body weight gain and obesity? BMC Med. 2022, 20, 1–9.
However, due to the added content in the introduction and discussion sections, three new references were added
Laffitte, A.; Neiers, F.; Briand, L. Functional roles of the sweet taste receptor in oral and extraoral tissues. Curr Opin Clin Nutr Metab Care 2014, 17, 379–385.
Fournel, A.; Marlin, A.; Abot, A.; Pasquio, C.; Cirillo, C.; Cani, P.D.; Knauf, C. Glucosensing in the gastrointestinal tract: Impact on glucose metabolism. Am. J. Physiol. Liver Physiol. 2016, 310, G645–G658.
Lee, G.; Chung, Y.-J.; Lee, M. Development of a gene expression panel, for the prediction of protein abundances in cancer cell lines. Curr. Bioinform. 2021, 16, 846–854.
Author must include the DOI for all references.
Response: This feedback was noted. However, the “instruction to Authors” section of the animals journal indicates that references should not contain the DOI. Please see the screenshot from the web page provided below.
Throughout all the manuscript, English writing needs to be checked.
Response: The English writing has been checked and several changes made. Some examples are presented below.
The sentence in lines 29-32 was edited from “This study investigated the effect of flavours in the diet of lambs on preference and feed efficiency, and the effect of flavours on sweet taste receptor heterodimer taste receptor family 1 member 2 and 3 (T1R2+T1R3) and sodium-glucose linked transporter 1 (SGLT1) genes expression in the small intestines.” to “This study investigated the effect of dietary flavour supplements on preference, feed efficiency and expression of the sweet taste receptor family 1 members 2 and 3 (T1R2+T1R3), and sodium-glucose linked transporter 1 (SGLT1) genes in the lambs small intestines.”
Line 64 was edited from” Ruminants more readily eat unfamiliar feed when associated with a familiar flavour or odour and avoid the feed when presented with a novel odour, and prefer some flavours to others” to “Ruminants are reported to eat unfamiliar feed more willingly when they are associated with a familiar flavour or odour and avoid the feed when presented with a novel odour. Preference of some flavours to others is also reported”
Introduction section
Line 59-61: The author mentions “animals” but the one cited study only refers to pigs. “Pluske, J.R.; Hampson, D.J.; Williams, I.H. Factors influencing the structure and function of the small intestine in the weaned 545 pig: a review. Livest. Prod. Sci. 1997, 51, 215–236”
Response: This citation has been replaced with three other citations that refer to cattle, goats and sheep. Please see the new references listed below.
De Souza Teixeira, O.; Kuczynski da Rocha, M.; Mendes Paizano Alforma, A.; Silva Fernandes, V.; de Oliveira Feijó, J.; Nunes Corrêa, M.; Andrighetto Canozzi, M.E.; McManus, C.; Jardim Barcellos, J.O. Behavioural and physiological responses of male and female beef cattle to weaning at 30, 75 or 180 days of age. Appl. Anim. Behav. Sci. 2021, 240, 105339.
Abdelsattar, M.M.; Vargas-Bello-Pérez, E.; Zhuang, Y.; Fu, Y.; Zhang, N. Impact of dietary supplementation of β-hydroxybutyric acid on performance, nutrient digestibility, organ development and serum stress indicators in early-weaned goat kids. Anim. Nutr. 2022, 9, 16–22.
McCoard, S.A.; Cristobal-Carballo, O.; Knol, F.W.; Heiser, A.; Khan, M.A.; Hennes, N.; Johnstone, P.; Lewis, S.; Stevens, D.R. Impact of early weaning on small intestine, metabolic, immune and endocrine system development, growth and body composition in artificially reared lambs. J. Anim. Sci. 2020, 98, 1–11
Line 77-83: Better not to mention “young ruminants”, the study focuses only on lambs.
Response: The change has been made to replace “young ruminants” with “lambs”
“Therefore, this study aimed to characterize the effect of flavour compounds in the diet of lambs on preference and feed efficiency in intensive farming and to use the basic knowledge of cellular level to examine the relationship between flavours and absorption of energetic compounds in the small intestines. We hypothesized that lambs prefer some flavours over others, and supplementing feed with their preferred flavours after weaning increase feed acceptance, intestinal expression of sweet taste receptors and monosaccharides transporters that lead to increased feed efficiency.”
Materials and methods section.
In the section 2.1 Preference Study: please provide a reference for the protocol used. Also, I recommend specifying if the flavor additives are non-nutritive or are energetic compounds.
Response: A reference to the protocol used had been provided in section 2.1. We have also specified that the flavour additives were non-nutritive.
“The preference test was then conducted according to the modified protocol of Harper et al. [11] using 16 different non-nutritive commercial flavours”
Section 2.2. The title of section 2.2 should include sampling or biopsy of intestinal tissue.
Response: This suggestion has been taken on board and section 2.2 title now reads, “2.2. Feed intake, performance, nutrient apparent digestibility and biopsy sampling of intestinal tissue”
On two occasions (lines 128 and 136) it is mentioned that a crossover experiment was carried out with a 4 x 2 arrangement. Write in a simple way without duplicating information.
Response: This section was reorganized and edited to eliminate duplication and improve clarity. It now reads “To examine the effect of flavours on feed intake, weight gain, apparent feed digestibility, energy balance and relative gene expression in the small intestines, a metabolic experiment was conducted with eight lambs (4 males and 4 females) using the sucram and capsicum flavours. Sucram and capsicum were selected because their preference was similar to the control during the preference test, and they are reported to influence feed efficiency in mammals [21,22]. The lambs (one male and one female) were assigned to either sucram, capsicum, mix (sucram and capsicum at 1:1 ratio) or no flavour control supplemented in the feed. The feed consisted of total mixed ration, whole corn grain and pellets in 50:25:25 ratio (weight as is), respectively, formulated to ensure 15.7% CP and 2.68 Mcal/kg metabolisable energy (ME) on DM basis (Table 1). Every morning, feed and flavours were weighed and mixed thoroughly by hand for two minutes, partitioned into 12 equal meal portions and loaded into the automatic feeders. Feed quantities were adjusted every day to ensure at least 5% refusals.”
Please, provide a reference for the intestine tissue collection protocol.
Response: The intestine tissue collection protocol was developed in our lab by a qualified veterinary surgeon but is not published.
Section 2.4
Line 193 and 195: “NanoDrop ND-1000 spectrophotometer” and “qPCRBIO cDNA synthesis kit” please provide the manufacturer information.
Response: The “NanoDrop ND-1000 spectrophotometer” and “qPCRBIO cDNA synthesis kit” manufacturer information has been provided. “Total RNA concentration was determined using NanoDrop ND-1000 spectrophotometer (Thermo Fisher Scientific, Waltham, MA, USA) before dilution to 200 ng/µl. A 1.0 µg of the total RNA from each tissue was reverse transcribed to cDNA using qPCRBIO cDNA synthesis kit (PCR Biosystems Inc. Wayne, PA, USA) according to the manufacturer’s protocol in a T100™ Bio-Rad Instrument.”
Section 2.5
Statistical analysis: Is the initial BW of each period considered as a covariate in the analysis?. It is not mentioned how the effect of baseline weight was controlled for in the study.
Response: We concur with the reviewer that baseline weight would introduce bias in the analysis. This was controlled by expressing daily weight gain in grams per kilogram metabolic body weight (g/Kg BW0.75) instead of just grams or kilograms per day. This information has been added in the materials and methods section 2.2. “Sheep were weighed at the beginning and at the end of each period of the experiment to determine weight gain (Scales Galore NY, U.S.A) and daily weight gain expressed in grams per metabolic body weight (g/Kg BW0.75) to control for baseline weight variability between periods.”
Line 227: “The model included flavour, run (first or subsequent exposure to the flavour), 227 sex, flavour-by-run interaction and flavour-by-sex interaction as the fixed effects”. No random effect was considered in the model?
Response: We thak the reviewer for pointing this omission. We agree that random variability is important in this analysis and we had included random effects in the model as indicated in the screenshot figure below. The error has been corrected in the Statistical analysis section. “The model included flavour, run (first or subsequent exposure to the flavour), sex, flavour-by-run interaction and flavour-by-sex interaction as the fixed effects, while the individual lambs nested in sex, lamb-by-run and lamb-by-flavour interactions were included as the random effects.”
A screenshot of the statistical model fitted using the repeated measures in JMP Pro software.
Results Section
Line 241: Abovementioned, significant differences were considered p<0.05. Please correct “p>0.1” to “p>0.05”.
Response: The “p>0.1” has been corrected to “p>0.05”.
“The flavour preference data are represented in Figure 1 and Figure 2. The effect of run, sex and sex by flavour interactions on flavour preference were not significant (p > 0.05)”
I recommend that the results in Figures 1-5 be shown as the mean ± standard error, rather than the standard deviation. This will help to better visualize the significance.
Response: Figures 1 to 5 are now presented in mean ± standard error as indicated below.
Figure 1. Consumption of grain mixture flavoured with liquid aroma (A; p < 0.001), non-aroma (B; p = 0.020) or powder-aroma flavours (C; p < 0.001) by sheep offered free choice for 5 minutes. Means ± standard error with different letters (a-c) differ significantly (p ≤ 0.05).
Figure 2. Consumption of grain mixture flavoured with liquid non-aroma (A; p = 0.023) or powder-aroma flavours (B; p < 0.001) during the first (Run 1) or subsequent (Run 2) exposure presented as mean ± standard error.
Figure 3. Difference in daily dry matter (DM), organic matter (OM), crude protein (CP), neutral detergent fibre (NDF), acid detergent fibre (ADF), hemicellulose (Hem) and ether extract (EE) intake between females and males fed unflavoured diet (control) or diet flavoured with sucram, capsicum or both expressed in mean ± standard error.
Figure 4. Effect of flavour (A; p = 0.049) and sex (B; p = 0.024) on daily weight gain per metabolic body weight of sheep fed unflavoured diet (Control) or diet flavoured with sucram, capsicum or both (mix) expressed in mean ± standard error.
Figure 5. Effect of sex on GEI (gross energy intake), DE (digestible energy), FE (faecal energy), GasE (combustible gas energy) and HiE (heat increment energy) of females and males fed unflavoured diet or diet flavoured with sucram, capsicum or both. Data are expressed in mean ± standard error.
Discussion section
Line 370-372: I doubt the cited reference support the idea of this sentence (Reference 33).
Response: This reference has been replaced with the correct one.
Diaz-sanchez, S.; Souza, D.D.; Biswas, D.; Hanning, I. Botanical alternatives to antibiotics for use in organic poultry production. Poult. Sci. 2015, 94, 1419–1430.
Fattori, V.; Hohmann, M.; Rossaneis, A.; Pinho-Ribeiro, F.; Verri, W. Capsaicin: Current understanding of its mechanisms and therapy of pain and other pre-clinical and clinical uses. Molecules 2016, 21, 844.
Line 419-420: Non ruminal fermentation characteristics were evaluated in this study, better not to discuss/concluded this way.
Response: This sentence has been deleted. “A previous study reported an improvement of feed digestibility by capsaicin supplementation in avian species [31], and the authors attributed the improvement to the activation of the sympathetic nervous system, increased secretion of digestive enzymes and bile acids [53]”
Section 4.3 T1R2, T1R3 and SGLT1 Relative Gene Expression: I suggest discussing the metabolic implications of the increase found in the T1R3 gene expression, the author may provide a potential explanation for this improvement.
Response: An explanation for the T1R3 gene expression increase has been provided.
“The increase in T1R3 gene expression may indicate that capsaicin and sucram caused an interaction in the taste transduction pathway that gave an impression of high sweet tasting stimuli in the small intestines lumen. The T1R3 is reported to form a T1R3/T1R3 homodimer at high monosaccharides and disaccharides concentration to increase uptake [77]. However, the increase in T1R3 gene expression did not amount to an increase in the SGLT1 expression, which may explain the lack of difference in energy balance. Also, the mRNA abundance does not accurately predict the protein abundance in some cases [21], hence the increase in the T1R3 gene expression may have not increased the T1R3 protein abundance.”
Line 513-514: “This may increase glucose absorption in the intestine and thus increase metabolic energy efficiency for growth”. Better not to mention this idea in the conclusion (suits better in the discussion section).
Response: Line 513-514 “This may increase glucose absorption in the intestine and thus increase metabolic energy efficiency for growth.” was deleted.
Reviewer 2 Report
The paper contains valuable data. Results were properly reported, and the findings have been accurately discussed and compared with other published papers. For further improvement of the manuscript, it requires some modification.
P1,L16 = Simple Summary
Change “improve sensory” to be “improve the sensory”.
Change “is scarcity” to be “is a scarcity”.
Change “flavours” to be “flavour”.
Change “and effect” to be “and the effect”.
P1,L29 = Abstract
Please rewrite this sentence ”This study investigated the effect of flavours in the diet of lambs on preference and feed efficiency, and the effect of flavours on sweet taste receptor heterodimer taste receptor family 1 member 2 and 3 (T1R2+T1R3) and sodium-glucose linked transporter 1 (SGLT1) genes expression in the small intestines”.
Change “concentrated orange” to be “orange”.
P2,L54 = Introduction
The introduction needs to be entirely re-written. It is very vague, and does not give the reader the necessary context to understand why you used the treatments you did, and why you made the measurements that you did. Please add new references to introduce your subject area, and compare it with other published material.
P6,L238 = Results
Results explained adequately.
P12,L234 = Discussion
Discussion explained adequately.
P16,L541 = References
Some of the references are old, should be replaced.
Regards
Author Response
We thank the reviewer for providing feedback and suggestions that will help improve the quality of this manuscript. We have addressed all the concerns individually as described in the attached file.

Reviewer 3 Report
Thank you for the interesting article. I have a few comments on it:
- in materials and methods it is good to add also information about the age of animals,
- the description about the experiment with flavor preference is not fully understood. Did all animals have access to all samples? When was the control mixture administered?
- it is also good to mention in the methodology that in the second part of the experiment the animals were assigned to groups of 1 male and 1 female each
- line 299, instead of Figure 6. should be Figure 5.
- Figure 5., I think on the X-axis it is better to use the abbreviations of their expansion in the caption
- Figure 6. it is also good to add the full name "Capsicum"
- line 385, the methodology states that the animals received 12 equal portions per day, so wasn't the feed limited?
- line 604- 605, the name of the journal is missing: Animal and Veterinary Sciences
Author Response
We thank the reviewer for providing feedback and suggestions that will help improve the quality of this manuscript.
Reviewer 3
Comments and Suggestions for Authors
Thank you for the interesting article. I have a few comments on it:
Response: We thank the reviewer for the feedback and suggestions.
- in materials and methods it is good to add also information about the age of animals,
Response: The age of the lambs has been provided. “To evaluate flavour preference in sheep, eight five-months-old Israeli crossbred Assaf lambs (four females and four males) weighing 41.0 ± 4.8 kg were purchased from a commercial farm.”
- the description about the experiment with flavor preference is not fully understood. Did all animals have access to all samples? When was the control mixture administered?
Response: This information has been added in the materials and methods section 2.1. “An unflavoured grains mixture was used as the control, offered at a random time similar to the flavoured treatments. The treatments were replicated twice, once in two days, to prevent carry-over effects and all the lambs had access to all the flavours.”
- it is also good to mention in the methodology that in the second part of the experiment the animals were assigned to groups of 1 male and 1 female each
Response: This information has been provided in section 2.2. “The lambs (one male and one female) were assigned to either sucram, capsicum, mix (sucram and capsicum at 1:1 ratio) or no flavour control supplemented in the feed.”
- line 299, instead of Figure 6. should be Figure 5.
Response: The correction has been made from Figure 6 to Figure 5. “However, the daily GEI, DE, FE, GasE and HiE were higher in males compared to the females (Figure 5; p ≤ 0.040).”
- Figure 5., I think on the X-axis it is better to use the abbreviations of their expansion in the caption
Response: This suggestion was taken on board and the new graph is provided below.
Figure 5. Effect of sex on GEI (gross energy intake), DE (digestible energy), FE (faecal energy), GasE (combustible gas energy) and HiE (heat increment energy) of females and males fed unflavoured diet or diet flavoured with sucram, capsicum or both. Data are expressed in mean ± standard error.
- Figure 6. it is also good to add the full name "Capsicum"
Response: Caps has been edited to Capsicum.
Figure 6. Relative gene expression of taste receptor family 1 member 2 (A; T1R2; p = 0.193), taste receptor family 1 member 3 (B; T1R3; p = 0.005) and sodium-dependent glucose transporter isoform 1 (C; SGLT1; p = 0.378) in proximal jejunum of sheep fed unflavoured diet (control) or diet flavoured with sucram, capsicum or both (mix).
- line 385, the methodology states that the animals received 12 equal portions per day, so wasn't the feed limited?
Response: This error has been corrected by deleting “ad libitum”. “In our study, lambs were offered high-energy diets ad libitum.”
- line 604- 605, the name of the journal is missing: Animal and Veterinary Sciences
Response: The name of the journal has been added. “El-Tazi, S.M.A. Response of broiler chicken to diets containing different mixture powder levels of red pepper and black pepper as natural feed additive. Anim. Vet. Sci. 2014, 2, 81–86.”

Round 2
Reviewer 1 Report
The observations and recommendations of the first review were addressed